# In Vitro Evaluation of No-Carrier-Added Radiolabeled Cisplatin ([^189, 191^Pt]cisplatin) Emitting Auger Electrons

**DOI:** 10.3390/ijms22094622

**Published:** 2021-04-28

**Authors:** Honoka Obata, Atsushi B. Tsuji, Hitomi Sudo, Aya Sugyo, Katsuyuki Minegishi, Kotaro Nagatsu, Mikako Ogawa, Ming-Rong Zhang

**Affiliations:** 1Department of Advanced Nuclear Medicine Sciences, Institute for Quantum Medical Science (iQMS), National Institutes for Quantum and Radiological Science and Technology (QST), 4-9-1 Anagawa, Inage-ku, Chiba 263-8555, Japan; obata.honoka@qst.go.jp (H.O.); minegishi.katsuyuki@qst.go.jp (K.M.); nagatsu.kotaro@qst.go.jp (K.N.); zhang.ming-rong@qst.go.jp (M.-R.Z.); 2Department of Molecular Imaging and Theranostics, Institute for Quantum Medical Science (iQMS), National Institutes for Quantum and Radiological Science and Technology (QST), 4-9-1 Anagawa, Inage-ku, Chiba 263-8555, Japan; sudo.hitomi@qst.go.jp (H.S.); sugyo.aya@qst.go.jp (A.S.); 3Graduate School of Pharmaceutical Sciences, Hokkaido University, Kita-ku, Sapporo, Hokkaido 060-0812, Japan; mogawa@pharm.hokudai.ac.jp; 4Japan Society for the Promotion of Science (JSPS), 5-3-1 Kojimachi, Chiyoda-ku, Tokyo 102-0083, Japan

**Keywords:** Auger electron, cisplatin, ^191^Pt, ^189^Pt, radio-drug, DNA double-strand break, γH2AX

## Abstract

Due to their short-range (2–500 nm), Auger electrons (Auger *e^−^*) have the potential to induce nano-scale physiochemical damage to biomolecules. Although DNA is the primary target of Auger *e*^−^, it remains challenging to maximize the interaction between Auger *e*^−^ and DNA. To assess the DNA-damaging effect of Auger *e*^−^ released as close as possible to DNA without chemical damage, we radio-synthesized no-carrier-added (n.c.a.) [^189, 191^Pt]cisplatin and evaluated both its in vitro properties and DNA-damaging effect. Cellular uptake, intracellular distribution, and DNA binding were investigated, and DNA double-strand breaks (DSBs) were evaluated by immunofluorescence staining of γH2AX and gel electrophoresis of plasmid DNA. Approximately 20% of intracellular radio-Pt was in a nucleus, and about 2% of intra-nucleus radio-Pt bound to DNA, although uptake of n.c.a. radio-cisplatin was low (0.6% incubated dose after 25-h incubation), resulting in the frequency of cells with γH2AX foci was low (1%). Nevertheless, some cells treated with radio-cisplatin had γH2AX aggregates unlike non-radioactive cisplatin. These findings suggest n.c.a. radio-cisplatin binding to DNA causes severe DSBs by the release of Auger *e*^−^ very close to DNA without chemical damage by carriers. Efficient radio-drug delivery to DNA is necessary for successful clinical application of Auger *e*^−^.

## 1. Introduction

Radiation is well known to induce DNA damage, leading to cell death or gene mutations, and its biological effects are widely used in diverse scientific fields [1]. Among the types of ionizing radiation, the Auger electron (Auger *e*^−^) has the shortest range (2–500 nm), yielding a high linear energy transfer (LET) of 4–26 keV/µm in the limited nano-scale range [2]. Multiple low-energy Auger *e*^−^ are released from an excited atom, and the locally absorbed radiation dose around a decaying site is ~1.6 MGy within a radius of 2 nm [3]. Because the effective range of Auger *e*^−^ is much smaller than the cell diameter (10–30 μm), Auger *e*^−^ has the potential to induce nano-scale physiochemical damage to targeted biomolecules by transporting Auger *e*^−^ emitters to specific intracellular regions.

There have been many attempts to develop substances radiolabeled with Auger *e*^−^–emitting radioelements, especially ^125^I and ^111^In [4], in order to apply Auger *e*^−^ to radiation therapy with radiopharmaceuticals (e.g., ^111^In-DTPA-octreotide, ^111^In-DTPA-EGFR, ^125^I-UdR, and ^125^I-huA33 [5,6,7,8,9,10]), but development is still ongoing. DNA is the primary target of Auger *e*^−^ to induce fatal DNA damage and cell death, as the range of Auger *e*^−^ is equivalent to the diameter of genomic DNA strands (~2 nm) [11,12,13,14,15]. Auger *e*^−^ emitters closer to the DNA induce more double-strand breaks (DSBs) [16,17,18], suggesting that delivery as close as possible to DNA is essential in order to maximize interaction between Auger *e*^−^ and DNA in the nano-scale range. Therefore, to achieve the maximum DNA-damaging effect of Auger *e*^−^, it is ideal for radioelements to directly bind to DNA on their own. Most radioelements cannot directly bind to DNA and must be used to label intermediary DNA-targeting molecules that are transported to DNA [19,20,21]; this inevitably creates separation between the DNA and the Auger *e*^−^ emitter.

Considering the above, here we focus on the platinum(Pt)-based antineoplastic drug cisplatin because it can form direct DNA adducts between Pt and nucleobases [22]. Therefore, cisplatin labeled with radio-Pt allows all Auger *e*^−^ to be released as close as possible to DNA. Unlike other radioelements, radio-Pt has a unique advantage that it can directly target DNA on its own, without intermediary DNA-binding compounds. Representative radio-Pt isotopes that emit Auger *e*^−^ are ^189^Pt (*T*_1/2_ = 10.87 h, EC), ^191^Pt (*T*_1/2_ = 2.83 d, EC), ^193m^Pt (*T*_1/2_ = 4.33 d, IT), and ^195m^Pt (*T*_1/2_ = 4.01 d, IT) [23,24], but the available isotopes are commonly produced in a reactor, resulting in carrier-added (c.a.) radio-Pt with low molar activity (~MBq/mg) [3,25,26]. Although the cytotoxicity of c.a. radio-cisplatin has been demonstrated in several previous studies [27,28,29], the effect of Auger *e*^−^ alone is masked by the chemotherapeutic effects of a large amount of non-radioactive cisplatin carriers. Consequently, biological studies using no-carrier-added (n.c.a.) radio-cisplatin are of considerable significance.

Recently, we established a procedure for producing n.c.a. [^189, 191^Pt]cisplatin (described as radio-cisplatin) [30,31]. The theoretical molar activity of the n.c.a. product would be on the scale of GBq/µg, making it possible to evaluate the potential effect of Auger *e*^−^ released from emitters bound to DNA with negligible chemical damage. In this study, we conducted in vitro experiments using n.c.a. radio-cisplatin. To investigate the intracellular behavior of n.c.a. radio-cisplatin at the picogram scale, cellular uptake, intracellular distribution, and DNA binding were compared with those of non-radioactive cisplatin at the standard dose (microgram scale) in chemotherapy. DSBs induced by n.c.a. radio-cisplatin emitting Auger *e*^−^ were evaluated by immunofluorescence staining of γH2AX in cell nuclei and electrophoresis of plasmid DNA to reveal the specific DNA-damaging effects of Auger *e*^−^.

## 2. Results

### 2.1. In Vitro Property of n.c.a. Radio-Cisplatin

#### 2.1.1. Cellular Uptake

Cellular uptake in two different tumor cell lines, H226 and LN319, was measured at seven time points from 30 min to 25 h (Figure 1). The uptake rate of n.c.a. radio-cisplatin was less than 1% of the incubated dose (ID) (Figure 1), as in other cell lines (data are not shown). This is probably because hydrophilic cisplatin does not efficiently pass through the cell membrane in any of these cell types. H226 and LN319 were used in the following experiments. Uptake by both cells increased with time, but the maximum uptake of n.c.a. radio-cisplatin was 0.56% ID in H226 and 0.57% ID in LN319 after 25-h incubation in FBS-free medium (Figure 1). In medium containing 10% FBS, the uptake rate of n.c.a. radio-cisplatin was similarly low, <1% ID (data are not shown). Thus, FBS has no significant effect on uptake. Comparison of uptake between c.a. and n.c.a. radio-cisplatin revealed differences in carrier dependency in the two tumor cells (Figure 1). In H226, uptake did not differ significantly between n.c.a. and c.a. radio-cisplatin, whereas in LN319, uptake of c.a. radio-cisplatin for LN319 was approximately twice as high as for radio-cisplatin after incubation for 5 h or more (Figure 1).

#### 2.1.2. Intracellular Distribution and DNA Binding

We investigated the intracellular distribution of radio-cisplatin by dividing cells into four subcellular fractions: F1, cytosolic; F2, membrane/organelle protein; F3, nuclear protein; F4, cytoskeletal fraction. Figure 2 (left) shows the intracellular distribution in H226 and LN319 cells, which were washed with PBS after incubation with radio-cisplatin in FBS-free medium for 3 h, and then incubated in fresh medium for 3 h or 17 h. Around 50–60% of intracellular radio-Pt was washed out into the fresh medium (data are not shown), and the residual radio-Pt in cells was fractionated. The F1 fraction increased, the F2 fraction was nearly constant, and the F3 and F4 fractions decreased from 3 h to 17 h (Figure 2 left). Figure 2 (right) shows the intracellular distribution in LN319 cells, which were incubated continuously with radio-cisplatin in FBS-medium without any medium change or washing. In Figure 2 (right), cellular uptake increased with time (data are not shown), in contrast to Figure 2 (left). No fractions showed a significant change.

Despite differences among the conditions and time points, the F3 fraction was almost constant, containing about 20% of the total radio-Pt in the cells regardless of the incubated radioactivity, indicating nuclear translocation of n.c.a. radio-cisplatin (Figure 2).

To determine to what extent radio-cisplatin in nuclei targets genomic DNA, we isolated genomic DNA from LN319 cells incubated with radio-Pt, and then measured its concentration and radioactivity. The DNA binding fraction of radio-Pt was 0.28 ± 0.02% ID/mg DNA. Based on the cellular uptake, we estimated that ~0.4% of intracellular radio-cisplatin (2% of intra-nucleus radio-cisplatin) bound to DNA.

Based on the intracellular distribution and DNA binding fraction, the number of radio-cisplatin per cell was roughly estimated assuming homogeneous uptake in individual cells: uptake of 0.1% ID of radio-cisplatin, intranuclear fraction of 0.02% ID (20% of intracellular radio-cisplatin), and DNA binding of 0.0004% ID (2% of intra-nucleus radio-cisplatin) (Table 1). When 3.5 × 10^10^ (1 × 10^5^ Bq) of [^191^Pt]cisplatin was added to 2 × 10^5^ cells (1.75 × 10^5^/cell), we estimated that 175 [^191^Pt]cisplatin molecules would be in each cell, 35 in each nucleus, and one bound to DNA (Table 1).

#### 2.1.3. Cytotoxicity

Toxicity in cells treated with n.c.a. radio-cisplatin, non-radioactive cisplatin, or saline only was evaluated by SRB cell proliferation assay and Live/Dead cell staining assay. As shown in Table 2 and Table 3, no cytotoxicity of n.c.a. radio-cisplatin was observed in either assay. In the cell proliferation assay of Table 2, there was no statistically significant difference between saline, non-radioactive cisplatin (1 ng), and radio-cisplatin. This is expected to be due to the low cellular uptake and the small DNA-binding fractions of n.c.a. radio-cisplatin (Figure 1 and Table 1).

### 2.2. DNA Damage Induced by n.c.a. Radio-Cisplatin

#### 2.2.1. Cellular Uptake Immunofluorescence Assay for γH2AX

As mentioned above, the low cellular uptake and the small DNA-binding fraction of n.c.a. radio-cisplatin prevented us from precisely evaluating cellular damage in a mass of cells. In order to detect low-probability DNA damage, it is necessary to analyze individual cells. Therefore, we conducted an immunohistostaining assay of γH2AX, as a reporter of DSBs to evaluate DNA damage by Auger *e*^−^ released very close to DNA. To assess direct DNA damage by Auger *e*^−^, DNA damage analysis was performed by gel electrophoresis of plasmid DNA.

Immunofluorescence assay for γH2AX was conducted in LN319 cells treated with n.c.a. radio-cisplatin (1200 kBq/^189^Pt + 600 kBq/^191^Pt), saline (untreated control), or non-radioactive cisplatin (7.5 ng) to compare with the chemical effect of cisplatin. Representative microphotographs are shown in Figure 3, and quantitative analysis is shown in Figure 4. We observed a 2-fold increase in γH2AX foci in nuclei treated with n.c.a. radio-cisplatin relative to those treated with saline or non-radioactive cisplatin. After treatment with n.c.a. radio-cisplatin, γH2AX signals were higher than in the other groups (Figure 3). Quantitative analysis of images revealed more γH2AX-positive nuclei (Figure 4 left) and 2-fold greater relative γH2AX intensity (Figure 4 right) in cells treated with radio-cisplatin than in the other groups. Although the difference was not statistically significant, radio-cisplatin tended to induce more γH2AX foci with greater fluorescence intensity than saline or non-radioactive cisplatin. To perform image analysis of DSBs induced by Auger *e*^−^, we obtained high-resolution two-dimensional (2D) and three-dimensional (3D) images with a 100 × objective lens (Figure 5). The 2D images indicated γH2AX aggregation in nuclei treated with radio-cisplatin (Figure 5 upper and middle panels), and the 3D images revealed their spatial dispersion (Figure 5 lower panes). These γH2AX aggregates with high fluorescence intensity corresponded to the quantitative results described above.

#### 2.2.2. Gel Electrophoresis of Plasmid DNA

To explore the possibility that Auger *e*^−^ directly induced the DSBs observed in the γH2AX assay, we conducted gel electrophoresis of plasmid DNA treated with radio-cisplatin (Figure 6). The original plasmid we purchased included circular and supercoiled forms, and the purified plasmid was only supercoiled. The restriction enzyme *Hin*dIII cuts one site in the plasmid, yielding the linear form. DMSO was used as an OH-radical scavenger to determine the effect of radicals.

Treatment with radio-cisplatin generated circular and linear forms (Figure 6, lane 5). Formation of the circular form was inhibited by DMSO (Figure 6, lane 8). Radio-HPLC analysis of plasmid and autoradiography of the gel revealed almost no DNA binding to n.c.a. [^191^Pt]cisplatin on day 1 (data are not shown). Due to the slow binding between Pt and DNA, most DNA damage (circular form) was likely due to radicals induced by radio-Pt. On the other hand, the band corresponding to the linear plasmid (direct DNA damage) was observed at the same intensity ratio both in the presence and absence of DMSO (Figure 6, lanes 5 and 8), suggesting that radio-cisplatin also caused direct plasmid breakage, probably by Auger *e*^−^ released from radio-Pt binding to DNA.

## 3. Discussion

This is the first study to evaluate the intracellular behavior and DNA-damaging effect of n.c.a. radio-cisplatin. Due to its low cellular uptake, only a small fraction of radio-cisplatin bound to DNA, inducing a low cytotoxic effect. On the other hand, radio-cisplatin induced severe DNA damage to a greater extent than non-radioactive cisplatin. Our findings reveal that the Auger *e*^−^ from n.c.a. radio-cisplatin directly induces severe DNA damage with a negligible chemical effect. To maximize the effect of Auger *e*^−^ in clinical applications, it will be necessary to develop a more efficient drug delivery system for radio-Pt.

### 3.1. In Vitro Property of n.c.a. Radio-Cisplatin

The uptake of n.c.a. radio-cisplatin increased with time, reaching a maximum of 0.6% at 25 h in both H226 and LN319 cells (Figure 1). The uptake and its time dependency were similar between the two cell lines, but the carrier dependency was different. There are likely to be two pathways of cellular uptake of cisplatin: passive diffusion through the plasma membrane and active transport mediated by membrane proteins [22,32,33]. This supports the idea that the uptake of cisplatin is time- and dose-dependent and is proportional to the administered concentration of cisplatin [34,35,36,37]. In the present study, the time dependency was observed in both H226 and LN319 cells under the n.c.a. condition (Figure 1), but the concentration dependency differed somewhat between the two cell lines. The cisplatin concentration was 70 pmol/L for n.c.a. and 33 µmol/L for c.a., a 10^6^-fold difference. The higher uptake in LN319 cells under the c.a. condition is consistent with previous studies [34,35,36,37]. The uptake in H226 cells, by contrast, did not differ significantly between the two conditions (Figure 1). Although the mechanism is unclear, differences in the activity of related active transporters, e.g., copper transporters and organic cation transporters [38,39,40], may explain the differences in carrier dependency between the cell lines.

The intracellular distribution and DNA-binding fraction of radio-cisplatin indicated that n.c.a. radio-cisplatin was translocated into nuclei and bound to DNA. The intra-nuclear fraction of 20% was consistent with a previous experiment [41]. Another study reported that approximately 1% of intracellular Pt bound to DNA [42], a little higher than we observed but still consistent with our result (0.4%). Thus, the intra-nuclear and binding fractions of n.c.a. radio-cisplatin were almost the same as those for c.a., as reported previously [41,42], whereas the n.c.a. dose level (pmol/L) was 10^6^-fold lower than the standard c.a. level (µmol/L). Our results suggest that both reaction rates of translocation to nuclei and binding to DNA are independent of Pt concentration.

Based on the intracellular behavior of n.c.a. radio-cisplatin, and the estimated number of Pt per cell, few Pt atoms bound to DNA in each cell (Table 1). This number was obtained by assuming equal uptake in individual cells, as a previous single-cell ICP-MS of non-radioactive cisplatin observed almost identical uptake of the standard dose of cisplatin, i.e., the uptake does not depend on the cell cycle [43,44,45]. Almost all radio-Pt decays without binding to DNA (Table 1), and consequently exerts minimal cell damage and induces a low rate of cell death (Table 2 and Table 3). Meanwhile, a previous study [27] reported that the cytotoxicity of c.a. [^191^Pt]cisplatin containing much more non-radioactive carrier (>µg) was greater than that of non-radioactive cisplatin alone, whereas non-radioactive cisplatin significantly decreased cell survival. The present study revealed a low cytotoxic effect of n.c.a. radio-cisplatin. Taken together, these findings suggest that the cytotoxicity of c.a. [^191^Pt]cisplatin is mainly due to chemical damage by non-radioactive cisplatin. Our results obtained with n.c.a. radio-cisplatin revealed that, due to the small DNA-binding fraction, the radiation damage has only a small effect on cell survival.

### 3.2. DNA Damage Induced by n.c.a. Radio-Cisplatin

Representative microphotographs of γH2AX staining and its quantitative analysis revealed more γH2AX foci with stronger fluorescence in nuclei treated with n.c.a. radio-cisplatin than in those treated with saline or non-radioactive cisplatin (Figure 3 and Figure 4), indicating that n.c.a. radio-cisplatin induced more DSBs in nuclei. Although the frequency of γH2AX positivity was at most 1% of all nuclei (Figure 4), this is reasonable considering the amount of radio-Pt binding to DNA, as expected from the low DNA binding rate (0.0004% ID) (Table 1). According to this calculation, in 1 × 10^6^ nuclei, 7 × 10^4^ atoms of radio-Pt were expected to decay, while bound to DNA, i.e., the proportion of nuclei with one radio-Pt decay in DNA is 7%. Therefore, this suggests that only a small amount of radio-Pt binding to DNA can induce the formation of DSBs.

More cisplatin induces more DSBs at microgram doses [46], whereas we observed no increase in γH2AX intensity in cells treated with non-radioactive cisplatin at nanogram doses, 10^2^-fold higher than the dose of n.c.a. radio-cisplatin (pg). This means that the most DSBs in cells treated with n.c.a. radio-cisplatin would be due to the effects of radiation, rather than the chemical effects of platinum. High-resolution 2D and 3D images revealed local aggregation of γH2AX in the nucleus (Figure 5). These highly fluorescent γH2AX aggregates were specifically observed in cells treated with n.c.a. radio-cisplatin, which likely corresponds to DSBs induced by Auger *e*^−^ released very close to DNA. Gel electrophoresis of the plasmids also supports the idea that DSBs were formed directly by radio-cisplatin, as DNA breakage was observed at the same level in the presence and absence of OH scavenger (Figure 6). On the other hand, there was a low probability of direct plasmid breakage and a high probability of indirect damage (Figure 6). Auger *e*^−^ are expected to directly induce DNA damage when released close to DNA. The low probability would be due to slow binding to the plasmid, causing Auger *e*^−^ to be emitted prior to DNA binding. Covalent binding between Pt and DNA would be unlikely to proceed efficiently, due to the very low concentration of both plasmid and n.c.a. radio-cisplatin (1–3 fmol/µL). Due to the low rate of covalent binding, the amount of direct DNA breakage by n.c.a. radio-cisplatin would be small.

Collectively, our observations of the in vitro properties of n.c.a. radio-cisplatin and its DNA-damaging effects indicate that severe DSBs were induced by Auger *e*^−^ released from radio-Pt bound to DNA, but not by the chemical effect of non-radioactive platinum carriers. Unfortunately, the low cellular uptake and the low DNA binding fractions led to a low probability of DSBs, resulting in low cytotoxicity. For successful clinical application of Auger *e*^−^, it is necessary to develop a more efficient system for delivering radio-Pt to DNA without modifying the DNA-binding groups.

## 4. Materials and Methods

### 4.1. General

Chemicals and reagents were purchased from FUJIFILM Wako Pure Chemical (Osaka, Japan), Tokyo Chemical Industry (Tokyo, Japan), Kanto Chemical (Tokyo, Japan), Otsuka Pharmaceutical Factory (Tokyo, Japan), Furuya Metal (Tokyo, Japan), Hayashi Pure Chemical Industry (Osaka, Japan), Fuso Pharmaceutical Industry (Osaka, Japan), or Merck (Darmstadt, Germany). Milli-Q ultrapure water or diluted water was used for dilution in all experiments.

HPGe *γ*-ray spectrometry was performed to measure radioactivity of [^189, 191^Pt]cisplatin in saline before all in vitro experiments. The HPGe detector (EGC 15–185-R; Eurisys Measures, Strasbourg, France) was coupled with a 4096 multi-channel analyzer (RZMCA; Laboratory Equipment, Ibaraki, Japan) and calibrated using a mixed (^109^Cd, ^57^Co, ^139^Ce, ^51^Cr, ^85^Sr, ^137^Cs, ^54^Mn, ^88^Y, and ^60^Co) standard source (Japan Radioisotope Association, Tokyo, Japan). A gamma-counter (Wizard2 2-Detector Gamma Counter; PerkinElmer, Waltham, MA, USA) was used to measure radioactivity in biological samples.

### 4.2. Synthesis of Carrier-Free [^189, 191^Pt]cisplatin

^189, 191^Pt was produced via the ^nat^Ir(p, xn)^189, 191^Pt reaction in a 30 MeV proton beam for 2–3 h at a beam current of 10 μA at the NIRS-QST isochronous cyclotron AVF-930, as described previously [30,31]. We used mixed ^189, 191^Pt, described as radio-Pt because ^189^Pt (*T*_1/2_ = 10.87 h, EC) is co-produced along with ^191^Pt (*T*_1/2_ = 2.83 d, EC) from a natural Ir target. [^189, 191^Pt]cisplatin (described as radio-cisplatin) was radio-synthesized [^189, 191^Pt]PtCl_4_^2-^ and prepared in a saline solution; radiochemical purity at the end of synthesis (EOS) was 99 + %. [^189, 191^Pt]cisplatin solution (300–2300 kBq/mL, ^189^Pt; 160–1500 kBq/mL, ^191^Pt) was used in all experiments, and the radioactivity ratio of ^189^Pt/^191^Pt at EOS was 0/1–2/1, depending on the cooling time.

### 4.3. Cell Culture

The human mesothelioma cell line NCI-H226 (H226) was obtained from ATCC (Manassas, VA, USA), and the human glioma cell line LN319 was from AddexBio Technologies (San Diego, CA, USA). The cells were cultured at 37 °C in a humidified atmosphere containing 5% CO_2_ in medium containing 10% FBS (Thermo Fisher Scientific, Waltham, MA, USA). Medium was RPMI-1640 (FUJIFILM Wako Pure Chemical) for H226 and D-MEM (FUJIFILM Wako Pure Chemical) for LN319.

### 4.4. Cellular Uptake

H226 (4 × 10^4^ cells/well) and LN319 (1 × 10^5^ cells/well) cells were seeded onto a 24-well plate and incubated overnight at 37 °C in a humidified atmosphere containing 5% CO_2_. After the medium was removed, FBS-free medium containing n.c.a. or c.a. (2.5 μg of non-radioactive cisplatin) radio-cisplatin (medium/saline = 9/1, 31 kBq/^189^Pt + 25 kBq/^191^Pt.) was added to the cells. After incubating for 0.5–24 h, the cells were washed with PBS and dissolved in 0.1 M NaOH. The radioactivity was measured using a gamma counter, and the protein concentration was determined with the Bradford reagent (Quick Start Bradford Protein Assay kit; Bio-Rad Laboratories, Hercules, CA, USA) for calculation of the uptake value.

### 4.5. Intracellular Distribution

H226 (1.7 × 10^6^ cells/dish) and LN319 (3 × 10^6^ cells/dish) cells were seeded onto a 6-cm dish and incubated overnight at 37 °C in a humidified atmosphere containing 5% CO_2_. (A) After the medium was removed, FBS-free medium containing radio-cisplatin (medium/saline = 3/2, 200 kBq/^189^Pt + 140 kBq/^191^Pt) was added to the cells. After incubating for 3 h, the medium was replaced with the medium containing 10% FBS, and the cells were incubated for 3–17 h. (B) N.c.a. radio-cisplatin in saline (1.8 MBq/^189^Pt + 1.6 MBq/^191^Pt) was directly added to the cells in medium containing FBS, and the cells were incubated for 6–48 h.

In both conditions, the cells were washed with PBS and collected using a cell scraper. The cells were separated into four fractions: F1 (cytosolic fraction), F2 (membrane/organelle protein fraction), F3 (nucleic protein fraction), and F4 (cytoskeletal fraction), using a ProteoExtract^®^ Subcellular Proteome Extraction kit (Merck, Darmstadt, Germany), followed by radioactivity measurements conducted for each fraction.

### 4.6. DNA Binding

LN319 (6.3 × 10^5^ cells/well) cells were seeded onto a 6-well plate and incubated overnight at 37 °C in a humidified atmosphere containing 5% CO_2_. n.c.a. radio-cisplatin in saline (920 kBq/^189^Pt + 530 kBq/^191^Pt) was added to the cells. After incubating for 1 d, genomic DNA was isolated from the cells using a NucleoSpin Tissue DNA extraction kit (Takara Bio, Kusatsu, Japan). Radioactivity of the collected solution containing genomic DNA was measured using a gamma-counter, and quantitative analysis was performed using a Microvolume UV-Vis Spectrophotometer NanoDrop One (Thermo Fisher Scientific). The DNA binding rate was calculated based on the measured values.

### 4.7. Sulforhodamine B (SRB) Assay

H226 (2 × 10^3^ cells/well) and LN319 (3 × 10^3^ cells/well) cells were seeded onto a 96-well plate and incubated overnight at 37 °C in a humidified atmosphere containing 5% CO_2_. N.c.a. radio-cisplatin (58–120 kBq/^189^Pt + 34–70 kBq/^191^Pt), non-radioactive cisplatin (1 μg, 1 ng), or saline was added to the cells, and the cells were incubated for 2 d. After the medium was removed and the cells were washed with PBS, the cells were immobilized with 50% trichloroacetic acid and stained with Sulforhodamine B dye (In Vitro Toxicology Assay Kit, Sulforhodamine B based, TOX6; Merck). After solubilizing the cells with Tris Base solution, absorbance at 565 nm was measured using a plate reader (Synergy HTX Multi-Mode Reader; BioTek, Winooski, VT, USA).

### 4.8. Live/Dead Viability/Cytotoxicity Assay

LN319 (2 × 10^4^ cells/well) cells were seeded onto a 24-well plate and incubated overnight at 37 °C in a humidified atmosphere containing 5% CO_2_. n.c.a. radio-cisplatin (75 kBq/^189^Pt + 50 kBq/^191^Pt), non-radioactive cisplatin (2.5 ng), or saline was added to the cells, and the cells were incubated for 2 d. After the medium was removed and the cells were washed with PBS, reduced-serum medium (Opti-MEM; Thermo Fisher Scientific) containing calcein/ethidium homodimer-1 (LIVE/DEAD Viability/Cytotoxicity Kit for mammalian cells; Thermo Fisher Scientific) was added to stain the cells. After incubating for 30 min, fluorescence was measured on a microplate reader (Spectra Max M5; Molecular Devices, Tokyo, Japan).

### 4.9. Immunofluorescence Staining of γH2AX

LN319 (4 × 10^5^) cells were seeded on coverslips (BioCoat collagen I 22-mm coverslips; Corning, Corning, NY, USA) and incubated overnight at 37 °C in a humidified atmosphere containing 5% CO_2_. Saline containing n.c.a. radio-cisplatin (1200 kBq/^189^Pt + 600 kBq/^191^Pt), non-radioactive cisplatin (7.5 ng), or none was added to the cells, and the cells were incubated for 3 h. After the medium was removed and the cells were washed with PBS, the cells were immobilized with ice-cold MeOH, incubated in 0.25% Triton X-100/PBS for permeabilization, and then incubated in 1% BSA/PBST for blocking. Immunofluorescence staining was conducted using a gamma-H2A.X antibody (phospho S139; Abcam, Cambridge, UK) as the primary antibody, with Alexa Fluor 594 anti-rabbit IgG (A-11012; Thermo Fisher Scientific) as the secondary antibody. Coverslips were placed on glass slides using a mounting medium containing DAPI (Vector Laboratories, Burlingame, CA, USA). Fluorescence images were taken with a BZ-X800 fluorescence microscope (KEYENCE, Osaka, Japan) and analyzed using the BZ-Z800 Analyzer software (KEYENCE), under the same conditions for each. In image analysis, setting the lower limits of fluorescence intensity to remove small foci as background, a nucleus with γH2AX foci was defined as a γH2AX-positive nucleus. Total nuclei and γH2AX-positive nuclei were counted, and the fluorescence intensity of nuclear γH2AX foci was quantified.

### 4.10. Gel Electrophoresis of Plasmid DNA

pcDNA3.1/Hygro (5597 nucleotides) was obtained from Thermo Fisher Scientific. The plasmids were transformed into competent *Escherichia coli* (DH5α; TOYOBO, Osaka, Japan). Transformed *E. coli* were cultured and plasmid was purified using QIAprep Mini Kit (QIAGEN, Venlo, Netherlands). Isolated plasmids were dissolved in TE buffer (0.01 mol/L Tris and 0.0001 mol/L EDTA) at a concentration of 100–200 ng/µL and stored at −20 °C. Plasmid samples were prepared at a DNA concentration of 5 ng/µL in 300 μL (1.5 μg = 0.4 pmol) of 0.2 mol/L phosphate buffer solution with or without 0.2 mol/L DMSO. Samples contained no cisplatin (control), [^191^Pt]cisplatin (1.62 MBq = 0.9 pmol), or non-radioactive cisplatin (0.18 ng = 0.6 pmol). After mixing, plasmid samples were incubated at 37 °C for 1 h, and then stored at <10 °C for 34 d. Original plasmid, cloned plasmid, and plasmid digested with the restriction enzyme *Hin*dIII (New England Biolabs, Ipswich, MA, USA) were also prepared at the same concentration (5 ng/µL) in TE buffer. Ten microliters of sample solution (50 ng of plasmid) was mixed with 2 μL of loading dye (sucrose red) and loaded in a gel lane. The gel consisted of 0.8% agarose and 0.1% ethidium bromide (EtBr) solution (10 mg/mL; NIPPON GENE, Tokyo, Japan) in 0.5× TBE buffer (0.0445 mol/L Tris, 0.0445 mol/L borate, and 0.001 mol/L EDTA). DNA ladder (1 kb) was obtained from (New England Biolabs). The gels were run at 100 V for 80 min, and then scanned on an image analyzer (ImageQuant LAS 500; Cytiva, Tokyo, Japan).

### 4.11. Statistical Analysis

The data in cellular uptake and SRB assay are expressed as means ± standard deviation (SD). The data of cellular uptake in Figure 1 were evaluated by two-way ANOVA, and the data from the cell proliferation assay in Table 2 and γH2AX assay in Figure 4 were evaluated one-way ANOVA with multiple comparisons using the GraphPad Prism 9 software (ver. 9.0.2, GraphPad Software, San Diego, CA, USA). *p* < 0.05 was considered statistically significant. The error of DNA binding was calculated from both counting errors and error propagation.

## 5. Conclusions

This is the first report to evaluate the property of n.c.a. radio-cisplatin (<1 pmol), which has the potential for direct DNA targeting and Auger *e*^−^ emission with no effect from chemical damage. Larger numbers of nuclear γH2AX foci, of greater intensity, in cell nuclei and more direct DNA breakage were observed in plasmid DNA, following treatment with n.c.a. radio-cisplatin, but not non-radioactive cisplatin. Unfortunately, the fraction of n.c.a. radio-cisplatin that bound to DNA was small, resulting in a low probability of DSB induction. Our findings suggest that radio-cisplatin binding to DNA directly causes severe DSBs by release of Auger *e*^−^ close to DNA. Hence, it would be valuable to develop a more efficient drug delivery system for radio-Pt for use in clinical applications.

## Figures and Tables

**Figure 1 ijms-22-04622-f001:**
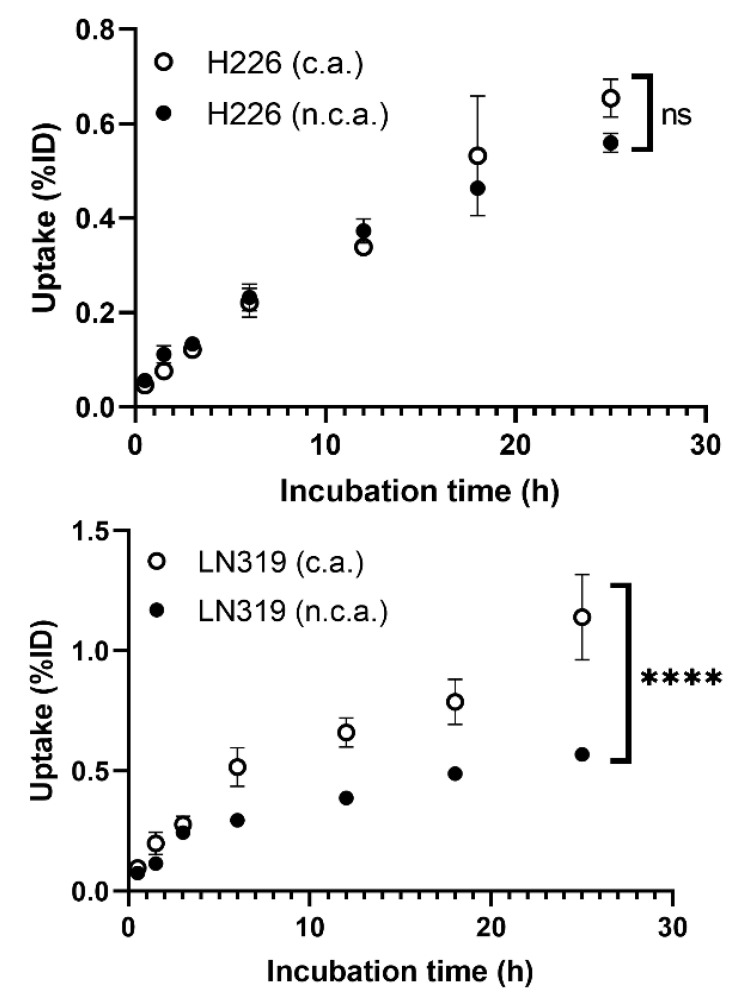
Cellular uptake of radio-cisplatin in H226 and LN319 cells. Incubated dose (ID) = 31 kBq/^189^Pt + 25 kBq/^191^Pt of radio-cisplatin with 0 (n.c.a.) or 2.5 µg (c.a.) of non-radioactive cisplatin. Data are expressed as means ± SD (*n* = 3). ns: not significant, **** *p* < 0.0001. H226 (4 × 10^4^ cells/well) and LN319 (1 × 10^5^ cells/well) cells were seeded 1 day before the assay.

**Figure 2 ijms-22-04622-f002:**
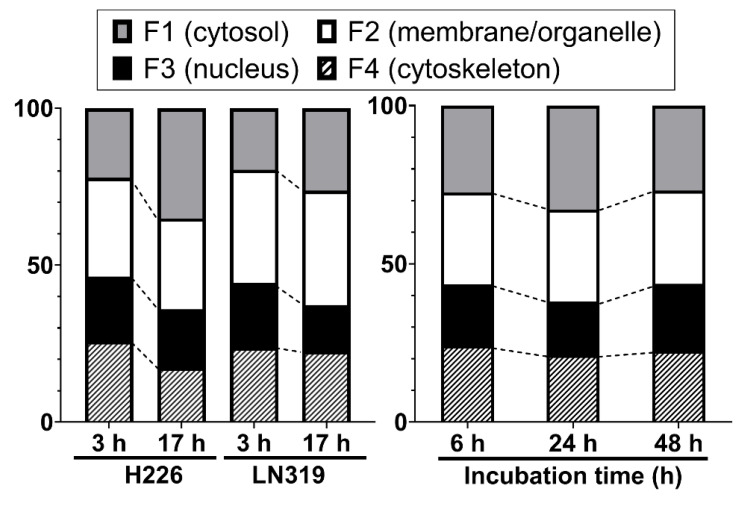
Intracellular distribution of radio-cisplatin in H226 and LN319 cells at 3 h and 17 h after incubation in FBS-free medium (**left**), and in LN319 after incubation for 6–48 h in medium containing FBS (**right**). F1, cytosolic fraction; F2, membrane/organelle protein fraction; F3: nucleic protein fraction; F4: cytoskeletal fraction.

**Figure 3 ijms-22-04622-f003:**
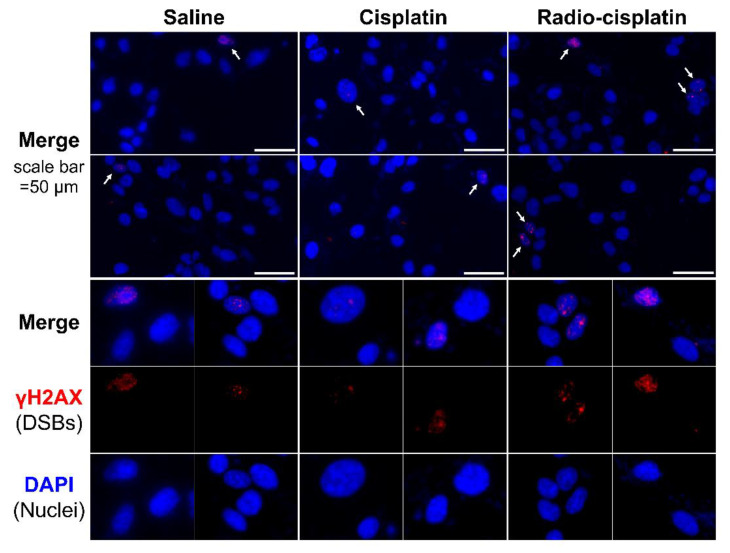
Representative microphotographs of γH2AX staining in LN319 cells treated with saline, non-radioactive cisplatin (7.5 ng), and n.c.a. radio-cisplatin (1200 kBq/^189^Pt + 600 kBq/^191^Pt). Cell nuclei are indicated in blue and γH2AX foci in red. Cells (4 × 10^5^/well) were seeded 1 day before the assay. Objective lens: 40×.

**Figure 4 ijms-22-04622-f004:**
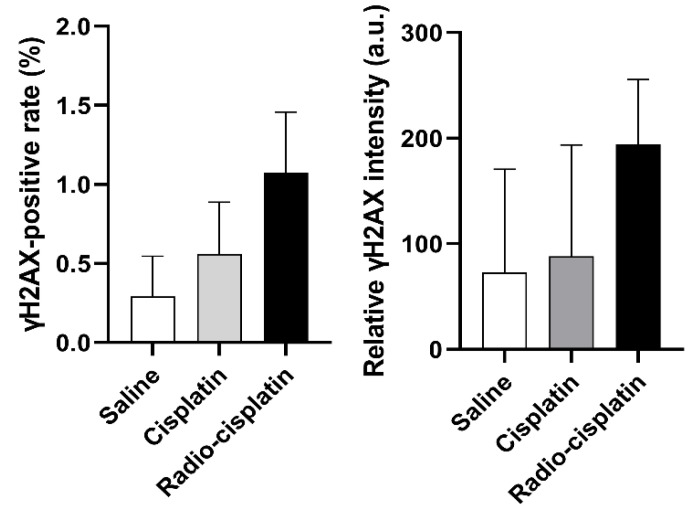
The fraction of LN319 cells with γH2AX-positive nuclei (**left**). Relative γH2AX intensity is expressed as the ratio of total fluorescence intensity of γH2AX in the nuclei to all nuclei counts (**right**). Data were obtained from three different images, including 199–274 nuclei per image, and expressed as means ± SD (*n* = 3). No significant difference was observed.

**Figure 5 ijms-22-04622-f005:**
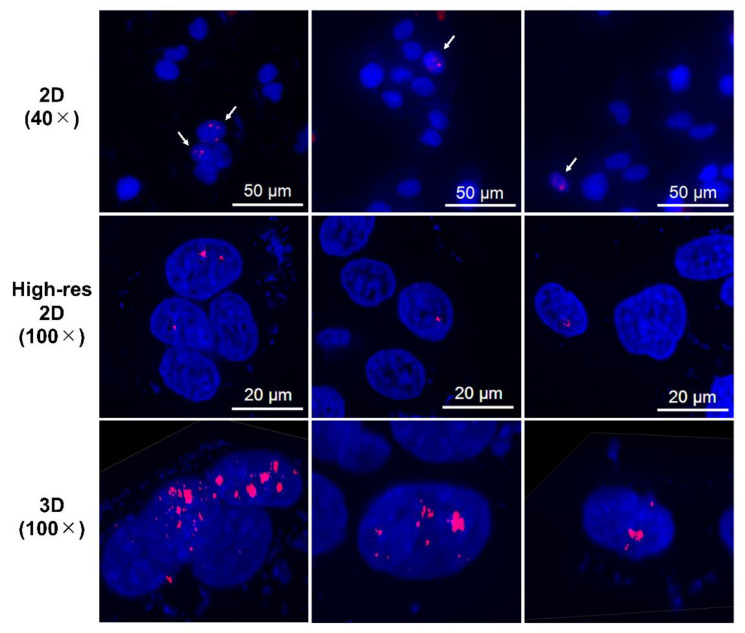
Representative high-resolution 2D and Z-stacked 3D images of cells with strong γH2AX foci (aggregations), observed in samples treated with n.c.a. radio-cisplatin. γH2AX-positive nuclei are indicated by arrows in the upper panels, and high-resolution 2D and 3D images are shown in the middle and lower panels.

**Figure 6 ijms-22-04622-f006:**
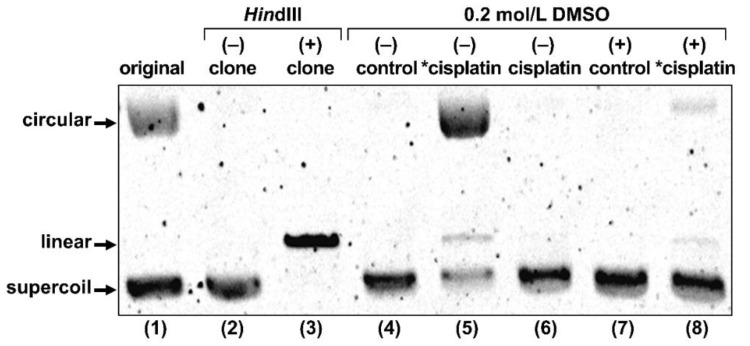
Gel electrophoresis of plasmid treated with no agent (control), radio-cisplatin (*cisplatin), or non-radioactive cisplatin (cisplatin). “original” refers to plasmid as purchased from the company. “clone” refers to our plasmid, purified after cultivation of the original clone. Each lane was loaded with 50 ng of plasmid in 10 μL of buffer.

**Table 1 ijms-22-04622-t001:** Estimated distribution of [^191^Pt]cisplatin in 2 × 10^5^ LN319 cells.

Fraction of ^191^Pt (% ID)		[^191^Pt]cisplatin	
		Number Per Cell	Radioactivity (Bq) Per Cell
Cell	0.1 *	175	5 × 10^−4^
Nucleus	0.02	35	1 × 10^−4^
DNA	0.0004	1	2 × 10^−6^

* Uptake at 3 h after incubation. ID: incubated dose = 1 × 10^5^ Bq.

**Table 2 ijms-22-04622-t002:** Cell proliferation SRB assay for H226 and LN319 cells.

Treatment	Survival (%) *	
	H226	LN319
Saline	100 ± 7	100 ± 11
Cisplatin		
1 ng (17 nM)	100 ± 9	95 ± 14
1 μg (17 µM)	63 ± 7	48 ± 8
[^189, 191^Pt]cisplatin		
^189^Pt, 58 kBq; ^191^Pt, 34 kBq	113 ± 18	92 ± 4
^189^Pt, 120 kBq; ^191^Pt, 70 kBq	91 ± 15	87 ± 13

* Survival is expressed as means ± SD (*n* = 6). There was no statistically significant difference.

**Table 3 ijms-22-04622-t003:** Fluorescence intensity in Live/Dead viability/cytotoxicity assay for LN319 cells.

Treatment	Live *	Dead *	D/L Ratio
Saline	900	6.77	1.00
Cisplatin	862	6.94	1.07
2.5 ng (11 nM)			
[^189, 191^Pt]cisplatin	829	7.27	1.17
^189^Pt, 75 kBq; ^191^Pt, 50 kBq			

* Fluorescence intensity (arbitrary units: a.u.).

## Data Availability

The data presented in this study are available on request from the corresponding author, upon reasonable request.

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
