# Peer review of "In Vitro Evaluation of No-Carrier-Added Radiolabeled Cisplatin ([189, 191Pt]cisplatin) Emitting Auger Electrons"

_ijms, 2021, doi:10.3390/ijms22094622_

Round 1

Reviewer 1 Report

In this manuscript, Obata et al., investigated an evaluation of radiolabeled cisplatin emitting augar elements and observation in detail on cancer cells,  the intracellular behavior of n.c.a. radio-cisplatin at the picogram scale, cellular uptake, distribution, cytotoxicity and DNA binding were compared with those of non-radioactive cisplatin. DNA damage induced by n.c.a. radio-cisplatin emitting Auger were analyzed by IF staining of γH2AX and electrophoresis of plasmid DNA to reveal the specific DNA- damaging effects of Auger e-. 

Suggestion to Author:  can further confirm DSBs by studying comet assay or PFGE. and also  cell cycle progression, apoptosis measurement chromosomal structural abnormality/genome instability need to be addressed.  

The aim of the study and outcome measures are clearly defined with appropriate references. Over all , this manuscript is a novel that will be of interest to the readers of the IJMS. 

Author Response

Suggestion to Author: can further confirm DSBs by studying comet assay or PFGE. and also cell cycle progression, apoptosis measurement chromosomal structural abnormality/genome instability need to be addressed. The aim of the study and outcome measures are clearly defined with appropriate references. Overall, this manuscript is a novel that will be of interest to the readers of the IJMS.

Answer: We deeply appreciate the kind comment and suggestion that encourage us for further investigations. Although the present submission focuses on the in-vitro behavior of n.c.a. radio-cisplatin and its potential DNA-damaging effect, a further study revealing the details is highly interesting. We will take the suggested experiments to our future works.

Reviewer 2 Report

The authors present the results on the studies of a procedure for producing no-carrier-added (n.c.a.) cisplatin and conducted in vitro experiments using n.c.a. radiolabeled cisplatin. To evaluate the property of n.c.a. radio-cisplatin, which has the potential for direct DNA targeting and Auger e- emission with no effect from chemical damage, the intracellular behavior of n.c.a. radio-cisplatin at the picogram scale, cellular uptake, intracellular distribution, and DNA binding were compared with those of non-radioactive cis-platin at the standard dose in chemotherapy. The text is very well written and hardly needs any improvements. The appropriateness of the literature citations is relevant. Summing up, the submitted manuscript entitled ‘In vitro evaluation of no-carrier-added radiolabeled cisplatin ([189, 191Pt]cisplatin) emitting Auger electrons’ authors: Honoka Obata, Atsushi B. Tsuji, Hitomi Sudo, Aya Sugyo, Katsuyuki Minegishi, Kotaro Nagatsu, Mikako Ogawa and Ming-Rong Zhang can be recommended for publication in Molecules. Please consider some minor changes to the text:

Line 16: It appears that ‘short range’ is missing a hyper. Please consider adding the hyphen: ‘short-range’

Line 28: The phrase ‘release’ seems to be missing a determiner before it. Please consider adding an article: ‘the release’.

Line 60: ‘the unique’ – Please consider making a change into ‘a change’

Line 85: It appears that ‘seven time’ is missing a hyper. Please consider adding the hyphen: ‘seven-time’

Line 109: Please consider changing ‘not shown’ into: ‘are not shown’

Line 124: Please consider changing the wording ‘In order to’ into: ‘To determine…’

Lines 207 and 208: Please consider changing the phase: ‘…it is likely that most of the DNA damage (circular form) was due…’ because it seems to be unclear and wordy.

Line 250: Please consider changing the phrase: ‘…there was independency on the cell cycle [43-45].’ because it seems to be unclear and wordy. The word ‘independency’ does not seem to fit this context.

Line 270: In the phrase, ‘formation’ seems to be missing a determiner before it. Please consider changing into: ‘the formation’.

Lines 353/354 and 372/373: Please consider replacing the word ‘over-night’ with ‘overnight’.

Author Response

We are grateful that the reviewer accepted the value of our manuscript and appreciate its useful comments for improving the paper.

Line 16: It appears that ‘short range’ is missing a hyper. Please consider adding the hyphen: ‘short-range’

Answer: We added the hyphen.

Line 28: The phrase ‘release’ seems to be missing a determiner before it. Please consider adding an article: ‘the release’.

Answer: The article has been added.

Line 60: ‘the unique’ – Please consider making a change into ‘a change’

Answer: We changed 'the' unique into 'a' unique.

Line 85: It appears that ‘seven time’ is missing a hyper. Please consider adding the hyphen: ‘seven-time’

Answer: The hyphen has been added.

Line 109: Please consider changing ‘not shown’ into: ‘are not shown’

Answer: We added 'are' between 'data' and 'not shown' over the manuscript.

Line 124: Please consider changing the wording ‘In order to’ into: ‘To determine…’

Answer: The ‘In order to’ has been into ‘To determine.'

Lines 207 and 208: Please consider changing the phase: ‘…it is likely that most of the DNA damage (circular form) was due…’ because it seems to be unclear and wordy.

Answer: We rephrased the sentence as '...most DNA damage (circular form) was likely due...'

Line 250: Please consider changing the phrase: ‘…there was independency on the cell cycle [43-45].’ because it seems to be unclear and wordy. The word ‘independency’ does not seem to fit this context.

Answer: We revised the ambiguous expression to '… the uptake does not depend on the cell cycle [43-45].'

Line 270: In the phrase, ‘formation’ seems to be missing a determiner before it. Please consider changing into: ‘the formation’.

Answer: The article has been added.

Lines 353/354 and 372/373: Please consider replacing the word ‘over-night’ with ‘overnight’.

Answer: Unfortunately, we cannot remove them due to the file format. Maybe the editorial office can do that.